# The Role of Vitamin D in Patients with Inflammatory Bowel Disease Treated with Vedolizumab

**DOI:** 10.3390/nu15224847

**Published:** 2023-11-20

**Authors:** Bincy P. Abraham, Christopher Fan, Theresa Thurston, Joshua Moskow, Hoda M. Malaty

**Affiliations:** 1Houston Methodist Gastroenterology Associates, Houston, TX 77030, USA; bpabraham@houstonmethodist.org (B.P.A.); cfan@houstonmethodist.org (C.F.); 2Weill Cornell Medical College, New York, NY 10065, USA; 3Houston Methodist Academic Institute, Houston, TX 77030, USA; 4School of Engineering Medicine, Texas A&M University, Houston, TX 77030, USA; theresa.thurston@tamu.edu (T.T.); jmoskow1997@tamu.edu (J.M.); 5Baylor College of Medicine, 7200 Cambridge St. Suite 10C, Houston, TX 77030, USA

**Keywords:** vitamin D, vedolizumab, Crohn’s disease, ulcerative colitis, inflammatory bowel disease, clinical outcomes

## Abstract

Background: Many clinical factors can contribute to the efficacy of medical therapy in Inflammatory Bowel Disease (IBD). We assessed their effects on the efficacy of vedolizumab therapy in a cohort of patients with IBD. Methods: We conducted a retrospective study on patients between 18 and 80 years of age with ulcerative colitis (UC) or Crohn’s disease (CD) who were seen in the IBD program at Houston Methodist in Houston, TX and treated with vedolizumab for at least 6 months from 2018 to 2022. We investigated factors prior to the initiation of therapy that best predicted treatment response, with an emphasis on vitamin D levels and examined several variables including patients’ demographics and clinical information on disease location and severity and nutritional status before and after the initiation of vedolizumab. Post-treatment data were gathered after a minimum of 6 months of vedolizumab therapy. The clinical parameters used for the study were the Harvey–Bradshaw Index for CD and the Activity Index for UC. Results: There were 88 patients included in our study of whom 44 had CD and 44 had UC.; median age was 39.5 (31.0, 53.25) years; 34% patients were male; and 80.7% were Caucasian. All patients received an induction dosing of 300 mg vedolizumab at 0, 2, and 6 weeks then maintenance dosing as standard of care every 8 weeks. Among UC patients with vitamin D ≥ 30 ng/mL at the initiation of vedolizumab therapy, UC Endoscopic Index of Severity (UCEIS) scores after 6 months of therapy were significantly lower than in those who had low pre-treatment vitamin D levels (1.5 vs. 3.87, *p* = 0.037). After treatment, vitamin D levels improved more significantly in the higher pre-treatment vitamin D group, with a median level of 56 ng/mL, than in the lower pre-treatment vitamin D group, with a median level of only 31 ng/mL (*p* = 0.007). In patients with CD with vitamin D ≥ 30 ng/mL at the initiation of vedolizumab therapy, we found higher iron saturation (12 vs. 25%, *p* = 0.008) and higher vitamin B12 levels (433.5 vs. 885 pg/mL, *p* = 0.003) than in those with vitamin D < 30 ng/mL. After treatment, CD patients with high pre-treatment vitamin D levels had significantly higher vedolizumab levels (27.35 vs. 14.35 μg/mL, *p* = 0.045) than those with low pre-treatment vitamin D. Post-treatment scores and inflammatory markers in CD patients (HBI, CRP, ESR, and SES-CD) were lower in those who had lower baseline vitamin D. Conclusions: Our results show higher pre-treatment vitamin D levels predicted significant endoscopic improvement in patients with ulcerative colitis (UC). Improving vitamin D levels lowered C-reactive protein levels significantly in CD patients. Higher vitamin D levels were seen after treatment in both UC and CD patients. Vitamin D can play a role in clinical and endoscopic outcomes and should be assessed routinely and optimized in patients with IBD.

## 1. Introduction

For patients with inflammatory bowel disease (IBD), including either ulcerative colitis (UC) and Crohn’s disease (CD), anti-integrin therapy is a reasonable treatment option after the failure of conventional biologic and small-molecule therapies. However, primary failure of response, incomplete response, and secondary loss of response to therapy contribute to increased symptoms and higher direct and indirect costs to healthcare [1]. Treatment failure may be due to alternate immune pathways not targeted by current mediations, low drug levels, antibody formation to a biologic, or suboptimal nutrition [2,3]. Ideally, we would be able to personalize the care for our patients by identifying factors that predict response to therapy, thus allowing us to choose the appropriate medication with the best efficacy.

Nutritional deficiencies in IBD have traditionally been considered a result of active inflammation with patients with IBD at risk for osteopenia and osteoporosis [4,5,6,7,8]. Patients with inflammatory bowel disease are known to have lower vitamin D levels at baseline, and those with surgeries are more at risk [9,10]. Moreover, recent reports have pointed to the importance of vitamin D in the natural history, treatment outcomes of IBD, and potentially all-cause mortality [11,12]. Indeed, previous studies have shown that low vitamin D levels are associated with disease severity and vedolizumab failure in patients with IBD [13,14,15]. Additionally, despite high doses of vitamin D, patients can achieve sufficiency even with supplementation [16,17]. We previously described, in a small group of patients, a lack of clinical remission when vitamin D levels are less than 30 ng/mL and persistent elevation in C-reactive protein (CRP) when vitamin D levels are extremely low (<15 ng/mL) [18]. Gubatan et al. recently showed that vitamin D levels less than 35 ng/mL predict a risk of relapse in UC patients with a sensitivity of 70% and a specificity of 74% [19]. Since evidence-based recommendations for vitamin D optimization in IBD are lacking, treating toward a level of at least 35 but optimally to 75 ng/mL appears safe and may benefit IBD disease activity [20]. In fact, polymorphisms in vitamin D receptors may confer differential risks for Crohn’s’ disease and ulcerative colitis [21]. However, a pre-diagnostic serum vitamin D level did not appear to be associated with the risk of UC or CD development [22].

Because vitamin D plays a role in modulating the immune system [23,24,25,26], we hypothesized that patients who have a high baseline vitamin D level of ≥30 ng/mL at the initiation of vedolizumab would have the best clinical and endoscopic responses to therapy. This vitamin D level is based on the classifications of normal, insufficient, and deficient levels of vitamin D [27]. 

We conducted the current study and analyzed vitamin D levels in patients with IBD on vedolizumab therapy to determine whether optimizing vitamin D levels can reduce inflammation and whether these levels can predict which patients with IBD respond best to treatment. 

## 2. Methods

This retrospective study included adult patients between 18 and 80 years of age with UC or CD who were seen at the Fondren IBD program at Houston Methodist in Houston, TX IBD program and treated with vedolizumab for at least 6 months from 2018 to 2022 and seen by our dedicated IBD providers: one IBD specialist and one IBD-trained physician associate. Patients with short bowel syndrome or on total parenteral nutrition were excluded from our analysis. 

We investigated the factors prior to the initiation of therapy that best predicted treatment response, with an emphasis on vitamin D levels. We examined several variables including patients’ demographics; clinical information on disease location and severity (i.e., C-reactive protein (CRP), erythrocyte sedimentation rate (ESR), fecal calprotectin, and lactoferrin); nutritional status (e.g., albumin, body mass index (BMI), vitamin D, iron, zinc, vitamin B12, and folate levels) before and after the initiation of vedolizumab; prior medications; and levels of vedolizumab therapy. Post-treatment data were gathered after a minimum of 6 months of vedolizumab therapy. The clinical parameters used for the study were the Harvey–Bradshaw Index for CD and the Activity Index for UC.

The exclusion criteria were patients diagnosed with indeterminate colitis or short bowel syndrome or on total parental nutrition and patients on vedolizumab less than 6 months or considering discontinuing vedolizumab within the next 4 months. We analyzed disease outcome after receiving vedolizumab therapy in IBD for a minimum of 6 months based on clinical, lab, and endoscopic assessments.

Our study aimed to determine whether optimizing vitamin D levels would improve clinical response, contribute to endoscopic improvement, and improve vedolizumab trough levels. We hypothesized that patients who have a high baseline vitamin D level of ≥30 ng/mL at the initiation of vedolizumab would have the best clinical and endoscopic responses to therapy. This vitamin D level is based on classifications of normal, insufficient, and deficient levels of vitamin D [27]. We analyzed disease outcomes after a minimum of 6 months of vedolizumab therapy for IBD based on clinical, lab, and endoscopic assessments. 

Continuous and categorical variables were described as means with standard deviations (SDs) or medians with interquartile ranges (IQRs) (Q1–Q3), respectively. Chi-square test was used for discrete variables, and the Mann–Whitney test was used to compare non-parametric continuous variables; otherwise, a student *t*-test was used for continuous variables. Statistical analyses were performed using RStudio (Boston, MA, USA). The study was conducted under a protocol approved by our institutional regulatory board.

## 3. Results

### 3.1. Patient Characteristics

From 2018 to 2022, 123 IBD patients were identified who had been on vedolizumab for over 6 months, but only 88 patients met our inclusion criteria with adequate follow-up data and were included in our study analysis (see Table 1). The remainder had missing labs, clinic visits, or endoscopy assessments, or they were lost to follow-up or returned to the referring provider. Half of the 88 patients included in our study population had been diagnosed with CD (*n* = 44), and the other half were diagnosed with UC (*n* = 44). The median age of the cohort was 39.5 (31.0, 53.25) years; 30 (34%) patients were male; and 71 (80.7%) were Caucasian. No statistically significant differences in age, gender, race, or number of prior therapies were seen between the UC and CD groups. In patients with CD, the location of the disease varied: most had colonic disease alone or colonic with small bowel disease. Most patients in both the UC and CD groups were on therapies prior to the initiation of vedolizumab with a median of one drug prior to initiation; CD patients had a median of one prior to surgery. Although the Short IBD Questionnaire was provided to all patients to complete, we had too few results (*n* = 11) to provide meaningful clinical or statistical significance. 

All patients received an induction dosing of 300 mg vedolizumab at 0, 2, and 6 weeks and then maintenance dosing as standard of care every 8 weeks, with some patients increased to every 6 weeks or every 4 weeks based on response and trough levels, as per clinical judgement.

### 3.2. Pre-Treatment Vitamin D Levels and Ulcerative Colitis Outcomes

Of the 44 patients that met our inclusion criteria, only 34 patients had pre-vedolizumab vitamin D levels (Table 2). In patients with UC with vitamin D ≥ 30 ng/mL at the initiation of vedolizumab therapy, we found that the UC Endoscopic Index of Severity (UCEIS) scores after 6 months of therapy were significantly lower than in those who had low pre-treatment vitamin D levels (1.5 vs. 3.87, *p* = 0.037). 

Although there was a trend toward lower ESR in the high vitamin D group, we found no statistically significant difference between groups in the clinical scores, CRP, or fecal markers prior to vedolizumab therapy and no difference in any of these values after treatment, regardless of vitamin D status. Albumin and BMI were not significantly associated with vitamin D levels either before or after treatment. Pre- and post-treatment nutritional levels of iron, vitamin B12, folate, and zinc were similar regardless of vitamin D levels. Although vedolizumab levels were numerically higher in the higher vitamin D group, this was not statistically significant. Only three patients with low vitamin D and four patients with high vitamin D required medication changes from vedolizumab. 

After treatment, the vitamin D levels improved more significantly in the higher pre-treatment vitamin D group, with a median level of 56 ng/mL, than in the lower pre-treatment vitamin D group, with a median level of only 31 ng/mL (*p* = 0.007). Though not statistically significant, UCAI, fecal calprotectin, and fecal lactoferrin were lower post-treatment in the low pre-treatment vitamin D group. The CRP, ESR, and UCEIS scores trended higher in those with low pre-treatment vitamin D. 

### 3.3. Pre-Treatment Vitamin D Levels and Crohn’s Disease Outcomes

Of the 44 patients that met our inclusion criteria, only 31 patients had pre-vedolizumab vitamin D levels (Table 3). In patients with CD with vitamin D ≥ 30 ng/mL at the initiation of vedolizumab therapy, we found higher iron saturation (12 vs. 25 %, *p* = 0.008) and higher vitamin B12 levels (433.5 vs. 885 pg/mL, *p* = 0.003) than in those with vitamin D < 30 ng/mL. Other nutritional factors, such as pre-treatment folate and zinc levels, did not significantly differ between the vitamin D groups. After treatment, CD patients with high pre-treatment vitamin D levels had significantly higher vedolizumab levels (27.35 vs. 14.35 μg/mL, *p* = 0.045) than those with low pre-treatment vitamin D. 

No differences in BMI and albumin were found between the vitamin D groups. After treatment, iron saturation improved more significantly in the higher than the lower vitamin D group (24 vs. 35.5 ng/mL, *p* = 0.007). Pre- and post-treatment clinical scores, inflammatory markers, endoscopic scores, and folate and zinc levels were similar regardless of the vitamin D levels. By the end of our study, only three patients with low vitamin D and three patients with higher vitamin D required medication changes in the CD patients. Post-treatment scores and inflammatory markers in CD (HBI, CRP, ESR, and SES-CD) were lower in those who had lower baseline vitamin D, though this was not statistically significant. 

### 3.4. Change in Vitamin D Levels and Outcomes

When we assessed the changes in the vitamin D levels, we saw a statistically significant drop in CRP among CD patients with baseline CRP > 5 mg/dL who had greater improvements in vitamin D (R^2^ = 0.3, *p* = 0.03); however, this did not hold true in patients with UC (R^2^=0.03, *p* = 0.589) (Figure 1a,b). None of the other parameters we analyzed—including clinical activity index scores (UCAI and HBI, Figure 2a,b) and endoscopy scores (UCEIS (UC Endoscopic Index of Severity), SES-CD (Simple Endoscopic Score for Crohn’s Disease), Figure 3a,b)—correlated with the changes in the vitamin D levels.

### 3.5. Pre- and Post-Vedolizumab Therapy Outcomes

#### Ulcerative Colitis Outcomes 

In patients with UC who received vedolizumab for 6 months, we found statistically significant decreases in the UC Activity Index from 6.0 [2.3, 8.0] to 1.0 [0.0, 2.5] (*p* < 0.001), and UCEIS scores from 6.0 [3.0, 8.8] to 1.0 [0.0, 5.0] (*p* < 0.001, Table 4). Vitamin D levels improved significantly from 33.2 [25.0, 38.8] to 45.0 [29.5, 60.0] (*p* = 0.008) ng/mL after vedolizumab initiation. Notably, although the CRP, ESR, calprotectin, and lactoferrin levels trended downward after the initiation of vedolizumab, these improvements were not statistically significant. Although nutritional labs and BMI improved with vedolizumab, these were not statistically significant.

### 3.6. Crohn’s Disease Outcomes

In patients with Crohn’s disease who received vedolizumab for 6 months, we found statistically significant improvements in the Harvey–Bradshaw Index (6.0 [3.0, 10.0] vs. 2.5 [1.00, 6.9], *p* = 0.002), SES-CD (15.0 [12.3, 23.3] vs. 3.0 [0.0 8.50], *p* = 0.002), and ESR (16.5 [6.8, 46.3] vs. 6.0 [2.0, 19.8] mm/hr, *p* = 0.004) after vedolizumab therapy (Table 5). As in UC, the vitamin D levels in patients with CD improved after the initiation of vedolizumab (23.0 [17.0, 36.0] vs. 31.0 [24.0, 51.7] ng/mL, *p* = 0.007). Iron (41.0 [25.3, 75.8] vs. 82.5 [62.0, 107.3] umol/L, *p* < 0.001) and iron sat levels (15.0 [9.0, 25.0] vs. 27.0% [21.8, 36.3], *p* < 0.001) also improved significantly with vedolizumab therapy. Nutritional labs and BMI also improved with vedolizumab, but these changes were not statistically significant.

## 4. Discussion

In our study of 88 patients with IBD on vedolizumab therapy for a minimum of 6 months, we found an improvement in the clinical activity scores and endoscopic scores in both our UC and CD patients that was consistent with the known efficacy of vedolizumab. In both the UC and CD patients, the vitamin D levels improved significantly after vedolizumab therapy. There are multiple potential mechanisms by which vedolizumab could increase levels of vitamin D, including a direct effect of the drug, a result of mucosal healing, decreased utilization of vitamin D for immune-mediated effects, or some other previously undescribed mechanism. 

In our patients with UC, although inflammatory blood and fecal markers, as well as clinical scores, did not differ between the vitamin D groups, we found significant improvements in the endoscopic scores in those with pre-treatment vitamin D levels ≥ 30 ng/mL. This would be consistent with previous publications showing that vitamin D levels less than 35 ng/mL predict a risk of relapse in UC and that vitamin D levels are associated with vedolizumab failure in patients with IBD [13,19]. If higher vitamin D levels can truly predict endoscopic improvement, then we should consider vitamin D optimization as a simple but effective co-management strategy to improve outcomes in our patients. However, inflammatory blood and fecal markers as well as clinical scores did not differ. The pre- and post-treatment nutritional levels of iron, vitamin B12, folate, and zinc did not differ between the low and high vitamin D groups. We suspect that this could be due to patients with UC having relatively normal values in general. 

In our patients with CD, we did not see any significant differences between the vitamin D groups in pre- and post-treatment clinical scores, inflammatory markers, or endoscopic scores. However, patients with higher pre-vedolizumab vitamin D levels (≥30 ng/mL) had higher iron saturation and vitamin B12 levels, suggesting concurring nutritional deficiencies in our CD patients. After treatment, iron saturation improved significantly more in the higher vitamin D group. Like our findings in the UC cohort, folate and zinc levels did not differ regardless of the vitamin D levels before and after vedolizumab treatment, likely due to patients having normal levels in general. 

In both diseases, albumin and BMI did not play a role regardless of the vitamin D levels, which suggests that the overall nutritional status using these clinical and lab assessments is not useful for predicting outcomes with vedolizumab therapy. This is important from the standpoint that vedolizumab is not a weight-based medication.

Interestingly, in the UC patient population, although post-treatment vedolizumab levels were numerically higher in the high vitamin D group, this difference was not statistically significant. However, in our CD population, vedolizumab drug levels were significantly higher in those with higher vitamin D levels. From a clinical standpoint, improving vitamin D levels may be a method to improve drug levels; however, vedolizumab may have indirectly improved vitamin D levels due to better absorption, mucosal healing, and general improvement in disease and nutritional status, as evidenced by overall significantly higher vitamin D levels after treatment in the high pre-treatment vitamin D group than in the low pre-treatment group. 

ESR and CRP were not significantly different in the low versus normal vitamin D level groups. However, we found when assessing whether changes in vitamin D levels correlated with clinical outcomes, that in patients with CD with a baseline pre-treatment CRP > 5 mg/dL, changes in vitamin D levels varied significantly with changes in CRP. In fact, patients with greater increases in vitamin D levels had a greater decrease in CRP. However, changes in vitamin D did not vary with other clinical measures of CD, such as HBI and SES-CD. This finding was unique to CD and was not seen in UC and may be in line with a systematic review and meta-analysis that showed that vitamin D supplementation can reduce the risk of clinical relapse in patients with IBD, specifically in CD [28]. However, previous reports have also suggested that vitamin D supplementation was associated with a reduction in intestinal inflammation [29] and could be related to disease differences or to small bowel absorption of vitamin D that may contribute to improvements in inflammation. However, a more recent Cochrane meta-analysis showed that there may be a signal for fewer clinical relapses when comparing vitamin D to placebo, but this was low-certainty evidence. Additionally, when comparing high-dose and low-dose vitamin D supplementation, there was no clear evidence of impact on the clinical response [30]. Moreover, pre-diagnostic serum vitamin D levels did not appear to be associated with the risk of UC or CD development [22].

Our study had several limitations. As this was a retrospective study, selection bias may play a role, as the providers chose appropriate patients for vedolizumab therapy. Unfortunately, laboratory values both pre- and post-vedolizumab were only available for a subset of patients, which can also introduce bias in the data. Additionally, given the limited number of patients included, this study was underpowered to show clear differences in outcomes. However, trends were apparent, more so in Crohn’s than ulcerative colitis. All patients in the current study were seen by the same IBD team (one physician and one physician assistant under the supervision of the same physician). All patients were treated with vitamin D supplementation if the levels were below 30 ng/mL as per standard of care. However, it is difficult to ascertain whether all patients adhered to the supplementation, and this may explain why some patients did not respond to oral vitamin D supplementation.

## 5. Conclusions

Our retrospective study of patients on vedolizumab therapy for both UC and CD found that higher pre-treatment vitamin D levels were associated with higher endoscopic improvement in UC and greater nutritional and supplementation improvements in iron saturation and B12 levels in CD. Those with higher vitamin D levels had higher vedolizumab drug levels in the CD population. In our CD patients, greater improvement in vitamin D levels correlated with lower CRP levels. This study suggests that vitamin D can play a role in clinical and endoscopic outcomes and should be assessed routinely and optimized in patients with IBD in a prospective study.

## Figures and Tables

**Figure 1 nutrients-15-04847-f001:**
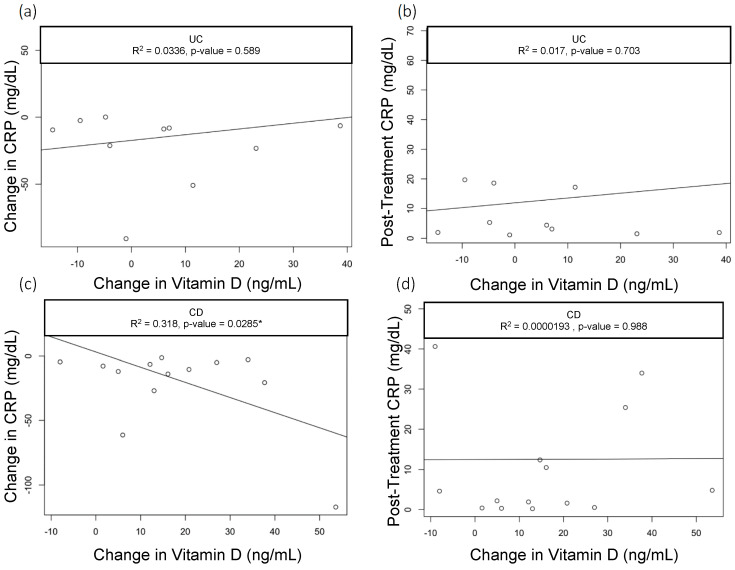
Correlation of Vitamin D with CRP in UC. Scatter plots with linear trend lines for UC: (**a**) change in CRP and change in vitamin D levels and (**b**) follow-up CRP levels and change in vitamin D levels. Scatter plots with linear trend lines for CD: (**c**) change in CRP and change in vitamin D nlevels and (**d**) follow-up CRP levels and change in vitamin D levels.

**Figure 2 nutrients-15-04847-f002:**
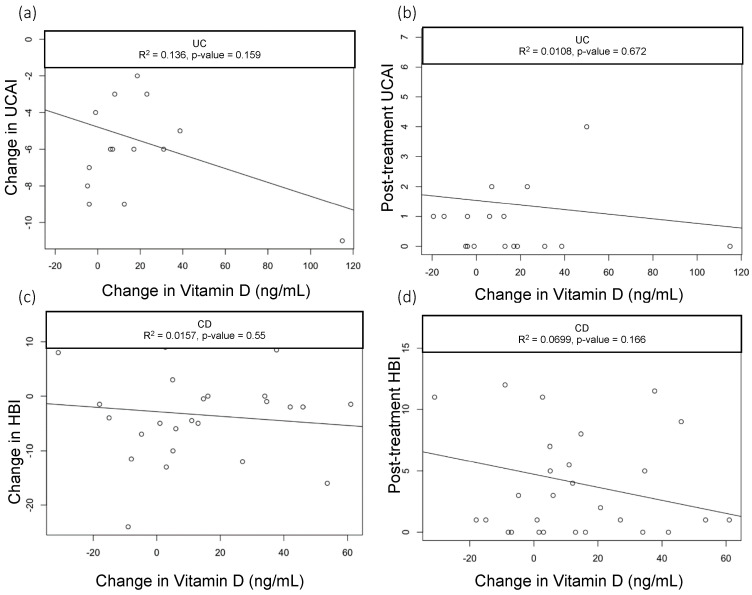
Correlation of UCAI scores with change in Vitamin D levels. Scatter plots with linear trend lines for UC: (**a**) change in UCAI and change in vitamin D levels and (**b**) follow-up UCAI score and change in vitamin D levels. Scatter plots with linear trend lines for CD: (**c**) change in HBI and change in vitamin D levels (**d**) and follow-up HBI score and change in vitamin D levels.

**Figure 3 nutrients-15-04847-f003:**
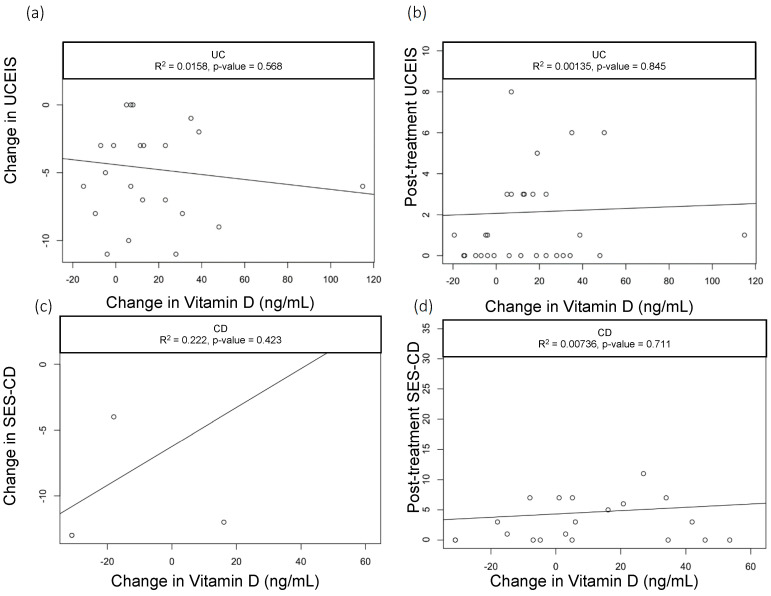
Correlation of UCEIS scores with change in Vitamin D levels. Scatter plots with linear trend lines for UC: (**a**) change in UCEIS and change in vitamin D levels and (**b**) follow-up UCEIS score and change in vitamin D levels. Scatter plots with linear trend lines for CD: (**c**) change in SES-CD and change in vitamin D levels and (**d**) follow-up SES-CD score and change in vitamin D levels.

**Table 1 nutrients-15-04847-t001:** Patient characteristics and clinical parameters used to assess vedolizumab outcomes.

	Combined (*n* = 88)	CD (*n* = 44)	UC (*n* = 44)	*p*-Value
**Age (median (IQR))**	39.5 [31.0, 53.3]	41.0 [29.0, 59.3]	38.00 [31.0, 50.3]	0.251
**Gender = Male (%)**	30 (34.1)	13 (29.5)	17 (38.6)	0.5
**Race (%)**				0.68
Asian	6 (6.8)	2 (4.5)	4 (9.1)	
Black	11 (12.5)	6 (13.6)	5 (11.4)	
Caucasian	71 (80.7)	36 (81.8)	35 (79.5)	
**Disease Duration (years)**	10.5 [5.1, 17.3]	11.5 [5.7, 21.8]	8.93 [5.1, 13.0]	0.069
**Location of Disease**				**<0.001**
Colonic	60 (68.2)	17 (38.6)	44 (100)	
Proximal	9 (10.2)	9 (20.5)	0 (0.0)	
Colonic + Proximal	19 (21.6)	18 (40.9)	0 (0.0)	
**# Prior Therapies**	1.0 [1.0, 2.0]	1.0 [1.0, 2.25]	1.0 [1.0, 2.0]	0.077
**# Prior Surgeries**	1.0 [1.0, 2.0]	1.0 [1.0, 2.0]	2.0 [1.5, 2.5]	0.487
**# Baseline Endoscopic Score** (UC = UCEIS, CD = SES-CD) (median [IQR])		15.00 [12.25, 23.25]	6.00 [3.00, 8.75]	N/A

Data shown in *n* (%) or median (IQR) unless otherwise stated. UCEIS: Ulcerative Colitis Endoscopic Index of Severity score. SES-CD: Simple Endoscopic Score for Crohn’s disease. Proximal includes small bowel +/− gastric disease. Bolded *p* value represents statistically significant findings.

**Table 2 nutrients-15-04847-t002:** Pre-vedolizumab vitamin D levels and Ulcerative Colitis outcomes.

	Vitamin D < 30 ng/mL (*n* = 16)	Vitamin D ≥ 30 ng/mL (*n* = 18)	*p*-Value
Age (years)	42.50 [33.25, 56.0]	39.50 [34.75, 48.0]	0.534
Male (%)	8 (50.0)	5 (27.8)	0.328
Race (%)			0.987
Asian	1 (6.2)	1 (5.6)	
Black	2 (12.5)	2 (11.1)	
Caucasian	13 (81.2)	15 (83.3)	
Disease Duration (years)	12.4 [6.2, 15.8]	9.4 [5.0, 13.1]	0.293
Vedolizumab Dose Escalation (n (%))	10 (62.5)	14 (77.8)	0.549
Iron Supplementation	1 (6.2)	1 (5.6)	1.00
Pre-Treatment Scores			
UCAI score	6.5 [4.0, 9.25]	6.0 [1.5, 8.0]	0.641
CRP (mg/dL)	4.0 [0.5 11.32]	1.5 [0.9, 5.2]	0.734
ESR (mm/h)	11.0 [6.0, 47.0]	6.0 [2.0, 11.0]	0.073
Fecal Calprotectin (μg/mg) (median (IQR))	299.0 [149.0, 844.0]	1377.9 [1313.9, 1441.8]	0.245
Fecal Lactoferrin (μg/g)	400.2 [400.2, 400.2]	71.7 [30.0, 130.9]	0.143
UCEIS Score (median (IQR))	7.0 [3.0, 9.0]	8.0 [4.5, 9.5]	0.716
BMI	24.6 [23.6, 29.1]	24.5 [22.1, 30.6]	0.72
**Vitamin D (ng/mL)**	**22.5 [17.5, 26.5]**	**38.7 [35.3, 48.6]**	**<0.001**
Albumin (g/dL)	3.7 [2.7, 3.9]	3.9 [3.1, 4.0]	0.533
Iron (umol/L)	62.0 [22.5, 100.5]	94.0 [58.0, 110.5]	0.11
TIBC (μg/dL)	311.0 [285.0, 361.0]	335.0 [293.5, 380.5]	0.42
Ferritin (μg/L)	38.0 [22.5, 115.5]	60.0 [29.5, 104.5]	0.575
Iron Sat (%)	20.5 [10.3, 28.0]	27.5 [21.0, 33.0]	0.164
Vitamin B12 (pg/mL)	526.5 [357.5, 825.3]	701.0 [414.3, 898.5]	0.378
Folate (ng/mL)	13.4 [10.6, 14.9]	12.2 [11.2, 18.1]	0.979
Zinc (μg/dL) (mean [SD])	68.7 (12.7)	82.7 (21.6)	0.173
# Prior Advanced Therapies (mean (SD))	1.00 [0.00, 1.00]	1.00 [1.00, 2.00]	0.199
Post-Treatment Scores			
UCAI core	0.5 [0.0, 3.3]	1.0 [0.0, 1.3]	0.838
CRP (mg/dL)	2.8 [1.1, 8.1]	2.1 [1.0, 4.6]	0.651
ESR (mm/h)	11.0 [5.8, 19.0]	6.0 [2.0, 11.0]	0.103
Fecal Calprotectin (μg/mg)	1026.0 [41.8, 2527.5]	1670.8 [1050.4, 2291.2]	0.643
Fecal Lactoferrin (μg/g)	37.0 [21.7, 58.8]	69.8 [50.2, 96.3]	0.564
**UCEIS score (mean (SD))**	**3.87 (3.7)**	**1.5 (2.5)**	**0.037**
Vedolizumab Level (μg/mL)	19.9 [11.4, 28.9]	25.4 [14.1, 34.5]	0.27
Vedolizumab Replacement Medication			0.459
Infliximab	1 (33.3)	2 (50.0)	
Ustekinumab	1 (33.3)	2 (50.0)	
Tofacitinib	1 (33.3)	0 (0.0)	
BMI	25.8 [24.3, 29.5]	24.8 [22.1, 28.8]	0.541
**Vitamin D (ng/mL)**	**31.0 [24.5, 49.5]**	**56.0 [46.0, 66.0]**	**0.007**
Albumin (g/dL)	3.8 [3.3, 4.0]	3.9 [3.7, 4.2]	0.338
Iron (umol/L)	79.5 [53.5, 87.0]	89.5 [54.3, 110.8]	0.351
TIBC (μg/dL)	328.0 [294.3, 360.3]	326.5 [299.3, 346.0]	1
Ferritin (μg/L)	54.0 [25.3, 122.5]	50.0 [25.0, 126.0]	0.885
Iron Sat (%)	24.50 [17.50, 33.00]	27.5 [17.3, 37.8]	0.809
Vitamin B12 (pg/mL)	556.0 [462.0, 606.5]	548.0 [469.0, 1093.0]	0.571
Folate (ng/mL)	12.6 [10.4, 18.5]	10.5 [7.5, 20.0]	0.321
Zinc (μg/dL) (mean (SD))	83.53 (9.99)	73.00 (5.66)	0.253

Data shown in *n* (%) or median (IQR (interquartile range) unless otherwise stated.

**Table 3 nutrients-15-04847-t003:** Pre-vedolizumab vitamin D levels (<30 ng/mL) and Crohn’s Disease outcomes.

	Vitamin D < 30 ng/mL (*n* = 19)	Vitamin D ≥ 30 ng/mL (*n* = 12)	*p*-Value
Age (years)	41.0 [30.5, 57.5]	43.5 [31.8, 62.0]	0.776
Male (%)	5 (26.3)	2 (16.7)	0.853
Race (%)			0.111
Asian	0 (0.0)	2 (16.7)	
Black	5 (26.3)	1 (8.3)	
Caucasian	14 (73.7)	9 (75.0)	
Disease Duration (years)	14.2 [4.5, 20.6]	14.4 [8.2, 25.4]	0.273
Disease Location (%)			0.339
1 Colon Only	8 (42.1)	3 (25.0)	
2 Proximal (Gastric ± Small bowel)	5 (26.3)	2 (16.7)	
3 Proximal and Colon	6 (31.6)	7 (58.3)	
Vedolizumab Dose Escalation (%)	8 (42.1)	8 (66.7)	0.335
Iron Supplementation	0 (0)	1 (8.3)	0.814
Pre-Treatment Scores			
HBI score	6.0 [3.0, 11.5]	5.0 [3.0, 10.0]	0.725
CRP (mg/dL)	8.2 [2.0, 21.8]	2.6 [0.6, 12.1]	0.212
ESR (mm/h)	25.0 [13.0, 50.0]	16.0 [5.0, 31.0]	0.403
Fecal Calprotectin (μg/mg)	N/A	272.5 [171.7, 373.2]	N/A
Fecal Lactoferrin (μg/g)	252.5 [252.5, 252.5]	52.5 [29.3, 75.7]	0.221
SES-CD score	17.0 [13.5, 22.0]	13.0 [11.5, 17.8]	0.721
# Prior Surgeries	1.0 [1.0, 1.0]	2.0 [1.0, 2.0]	0.129
BMI (kg/m^2^)	24.1 [19.3, 30.4]	22.4 [21.5, 24.2]	0.525
**Vitamin D (** **ng/mL)**	**17.4 [14.7, 22.0]**	**40.0 [33.8, 49.8]**	**<0.001**
Albumin (g/dL)	3.0 [2.5, 4.0]	3.6 [3.3, 4.1]	0.451
Iron (umol/L)	35.5 [25.0, 57.8]	81.0 [26.0, 90.5]	0.132
TIBC (μg/dL)	301.0 [266.5, 383.5]	323.0 [247.0, 327.0]	0.586
Ferritin (μg/L)	58.0 [14.0, 141.0]	82.0 [60.0, 122.0]	0.35
**Iron Sat (%)**	**12.0 [6.25, 17.25]**	**25.0 [17.5, 28.5]**	**0.008**
**Vitamin B12 (pg/mL)**	**433.5 [378.5, 551.0]**	**885.0 [645.5, 1285.3]**	**0.003**
Folate (ng/mL)	9.6 [6.8, 13.7]	18.6 [10.2, 20.7]	0.054
Zinc (μg/dL) (mean (SD))	66.9 (20.0)	73.8 (17.6)	0.521
Prior Biologics	1.00 [1.00, 2.00]	2.00 [1.00, 3.00]	0.13
Post-Treatment Scores			
HBI score	1.0 [0.0, 5.0]	5.5 [1.0, 10.0]	0.163
CRP (mg/dL)	1.6 [0.4, 4.3]	3.0 [1.5, 18.7]	0.101
ESR (mm/h)	7.5 [2.0, 21.3]	9.0 [2.0, 11.0]	0.692
Fecal Calprotectin (μg/mg)	24.0 [12.3, 702.0]	2.5 [2.5, 156.7]	0.825
Fecal Lactoferrin (μg/g)	194.0 [29.3, 737.3]	30.0 [30.0, 58.6]	0.793
SES-CD Score (mean (SD))	3.6 (3.8)	6.3 (12.9)	0.539
**Vedolizumab level (μg/mL)**	**14.4 [11.1, 23.0]**	**27.4 [22.4, 42.7]**	**0.045**
Vedo Replacement Drug			0.189
Infliximab	1 (33.3)	2 (66.7)	
Ustekinumab	2 (66.7)	0 (0.0)	
Tofacitinib	0 (0.0)	1 (33.3)	
BMI (kg/m^2^)	24.7 [21.6, 33.5]	24.4 [22.3, 26.5]	0.446
Vitamin D (ng/mL)	27.0 [22.0, 50.5]	31.0 [29.5, 64.4]	0.081
Albumin (g/dL)	3.5 [2.7, 3.9]	3.5 [3.3, 4.1]	0.602
**Iron (umol/L)**	**65.0 [54.0, 86.0]**	**96.5 [77.0, 116.3]**	**0.025**
TIBC (μg/dL)	305.0 [278.0, 374.0]	266.5 [255.5, 338.0]	0.335
Ferritin (μg/L)	83.0 [52.5, 300.0]	144.0 [61.5, 256.0]	0.621
**Iron Sat (%)**	**24.0 [17.0, 30.0]**	**35.5 [33.5, 40.8]**	**0.007**
Vitamin B12 (pg/mL)	572.0 [445.5, 718.5]	598.0 [473.0, 1243.0]	0.445
Folate (ng/mL)	10.7 [9.6, 19.2]	17.1 [10.4, 19.3]	0.631
Zinc (μg/dL) (mean (SD))	71.13 (10.59)	77.40 (13.56)	0.411

Data shown in *n* (%) or median (IQR) unless otherwise stated. Statistically significant results are bolded.

**Table 4 nutrients-15-04847-t004:** Ulcerative Colitis pre- and post-vedolizumab clinical, endoscopic, and laboratory values.

	Pre-Vedolizumab	Post-Vedolizumab	*p*
**UC Activity Index**	**6.0 [2.3, 8.0]**	**1.0 [0.0, 2.5]**	**<0.001**
CRP (mg/dL)	3.2 [0.9, 11.3]	2.5 [1.1, 6.7]	0.777
ESR (mm/h)	11.0 [6.0, 27.0]	7.5 [2.0, 16.8]	0.245
Fecal Calprotectin (μg/mg)	1250.0 [299.0, 1250.0]	196.0 [52.0, 2000.0]	0.479
Fecal Lactoferrin (μg/g)	130.9 [50.9, 276.1]	49.0 [28.2, 81.1]	0.134
**UCEIS**	**6.0 [3.0, 8.8]**	**1.0 [0.0, 5.0]**	**<0.001**
BMI (kg/m^2^)	24.4 [21.3, 28.7]	25.87 [23.1, 29.9]	0.337
**Vitamin D (ng/mL)**	**33.2 [25.0, 38.8]**	**45.0 [29.5, 60.0]**	**0.008**
Albumin (g/dL)	3.9 [3.0, 4.1]	3.9 [3.6, 4.2]	0.484
Iron (umol/L)	65.0 [34.0, 101.0]	83.0 [52.3, 102.3]	0.423
TIBC (μg/dL)	322.0 [283.5, 361.8]	324.5 [295.0 353.0]	0.563
Ferritin (μg/L)	54.5 [23.3, 135.0]	55.0 [26.0, 122.0]	0.766
Iron Sat (%)	27.0 [12.8, 29.0]	25.5 [16.8, 32.3]	0.477
Vitamin B12 (pg/mL)	619.0 [404.8, 840.8]	548.0 [440.0, 687.0]	0.637
Folate (ng/mL)	12.5 [10.7, 15.4]	11.3 [9.2, 19.9]	0.942
Zinc (μg/dL) (mean (SD))	77.4 (19.6)	79.9 (8.8)	0.741

Statistically significant findings noted in bold.

**Table 5 nutrients-15-04847-t005:** Crohn’s Disease pre- and post-vedolizumab clinical, endoscopic, and laboratory values.

	Pre-Vedolizumab	Post-Vedolizumab	*p*
**Harvey–Bradshaw Index**	**6.0 [3.0, 10.0]**	**2.5 [1.0, 6.9]**	**0.002**
CRP (mg/dL)	3.4 [0.9, 13.5]	2.7 [0.6, 6.7]	0.44
ESR (mm/h)	**16.5 [6.8, 46.3]**	**6.0 [2.00, 19.8]**	**0.004**
Fecal Calprotectin (μg/mg)	326.5 [152.0, 495.7]	80.5 [2.5, 235.3]	0.234
Fecal Lactoferrin (μg/g)	98.9 [52.5, 175.7]	30.0 [29.0, 96.2]	0.735
**SES-CD**	**15.0 [12.3, 23.3]**	**3.0 [0.0, 8.5]**	**0.002**
BMI (kg/m^2^)	22.7 [19.7, 28.0]	24.0 [21.6, 29.7]	0.154
**Vitamin D (ng/mL)**	**23.0 [17.0, 36.0]**	**31.0 [24.0, 51.7]**	**0.007**
Albumin (g/dL)	3.3 [2.6, 4.2]	3.5 [3.1, 4.1]	0.658
**Iron (umol/L)**	**41.0 [25.3, 75.8]**	**82.5 [62.0, 107.3]**	**<0.001**
TIBC (μg/dL)	297.0 [256.0, 330.5]	312.0 [270.0, 344.3]	0.305
Ferritin (μg/L)	74.0 [26.0, 141.0]	117.0 [54.0, 273.5]	0.064
**Iron Sat (%)**	**15.0 [9.0, 25.0]**	**27.0 [21.8, 36.3]**	**<0.001**
Vitamin B12 (pg/mL)	545.0 [407.0, 969.0]	572.0 [443.0, 946.3]	0.55
Folate (ng/mL)	11.0 [6.8, 18.0]	11.7 [8.9, 19.3]	0.462
Zinc (μg/dL) (mean (SD))	70.1 (18.5)	74.3 (11.7)	0.446

Statistically significant findings noted in bold.

## Data Availability

Data are contained within the article.

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
