# Peer review of "The Role of Vitamin D in Patients with Inflammatory Bowel Disease Treated with Vedolizumab"

_nutrients, 2023, doi:10.3390/nu15224847_

Round 1

Reviewer 1 Report

Comments and Suggestions for Authors

The manuscript presents interesting data on the possible role of vitamin D in the result of treatment with vedolizumab in patients with IBD. Based on a cohort of 88 patients (half with UC and half with CD), the authors analyzed vitamin D levels, among other parameters, before and after vedolizumab treatment. They correlated it with the result of therapy. These data could support controlling the vitamin D level before treatment to have a better response (as a co-treatment).

The paper is generally well written; still, some changes should be made.

The abstract must include more data on the cohort, the methods, the results (with data), and conclusions. It would be better to be structured.

The writing style may be verified to avoid repetition inside the same phrases and sentences (see lines 35-36).

The figures should be verified to have correct axis labels (see Fig 2).

The Discussion section should be rewritten to include a discussion of the results in the view of other data from the literature. Surprisingly, there is no reference cited in this section. This part must be for sure rewritten and improved.

Author Response

Reviewer-1

The manuscript presents interesting data on the possible role of vitamin D in the result of treatment with vedolizumab in patients with IBD. Based on a cohort of 88 patients (half with UC and half with CD), the authors analyzed vitamin D levels, among other parameters, before and after vedolizumab treatment. They correlated it with the result of therapy. These data could support controlling the vitamin D level before treatment to have a better response (as a co-treatment).

  1. The paper is generally well written; still, some changes should be made.

I did go through the editing the manuscript couple of times.

  1. The abstract must include more data on the cohort, the methods, the results (with data), and conclusions. It would be better to be structured.

I reconstructed the abstract and inserted it in the manuscript.

  1. The writing style may be verified to avoid repetition inside the same phrases and sentences (see lines 35-36).

I responded to repetition issue all over the text.

  1. The figures should be verified to have correct axis labels (see Fig 2).

Figure 2 has the correct axis labels. Change in activity index vs change in vitamin D levels

  1. The Discussion section should be rewritten to include a discussion of the results in the view of other data from the literature. Surprisingly, there is no reference cited in this section. This part must be for sure rewritten and improved.

We have edited the discussion and added citations to help show how our data can be understood in the context of already published data relating vitamin D and IBD.

Reviewer 2 Report

Comments and Suggestions for Authors

It is an interesting paper, and the authors mentioned the limitation of the study, which may affect the results beautifully and clearly, this is a good point to their credit, but I have some comments

1-      As long as this study was traced for six months, why were diarrhea and constipation not considered? Exposure to long-term VD may contribute directly to several digestive issues, which affect the absorption of drugs and others.

2-      Because of no close contact between the physician and patients, the irregular taking of VD might occur.

3-      Were levels of VD3 taken into account before the prescription of VD pre-therapy?

4-      As long as vedolizumab levels were numerically higher in the higher vitamin D group, the correlation between VD and integrin protein must be clearer.

5-      Calcium levels consideration is a little bit important because high calcium levels stimulate VD3 release.

6-      The authors mentioned the pre and post-therapy VD levels of the patients but did not mention them during the Vedolizumab therapy. 

Author Response

Reviewer-2

  1. As long as this study was traced for six months, why were diarrhea and constipation not considered? Exposure to long-term VD may contribute directly to several digestive issues, which affect the absorption of drugs and others.

Constipation was not a common symptom in any of our IBD patients and should not interfere with vitamin D absorption.

For those with diarrhea, especially with active disease, we took into account disease activity when accounting for the results.  This is exactly the point of the study that absorption could affect even vitamin D absorption from either diet or vitamin D oral replacement. 

  1. Because of no close contact between the physician and patients, the irregular taking of VD might occur.

This is true in that we were not directly observing patients taking vitamin D and that is a limitation of a retrospective; non direct observed therapy.  However most patients would notify us if they were appropriately taking their supplements or not and we took their history at face value.  This point is stated in the discussion section.

  1. Were levels of VD3 taken into account before the prescription of VD pre-therapy?

Yes, that is why we have pre-vitamin D levels and post -starting vedolizumab vitamin D levels. 

  1. As long as vedolizumab levels were numerically higher in the higher vitamin D group, the correlation between VD and integrin protein must be clearer.

Not sure what they mean by integrin protein?

  1. Calcium levels consideration is a little bit important because high calcium levels stimulate VD3 release.

(We did not find any significant abnormalities in calcium levels in our study.)

 6-      The authors mentioned the pre and post-therapy VD levels of the patients but did not mention them during the Vedolizumab therapy. 

Post therapy includes patients during vedolizumab therapy.  “Post” was defined as “after starting vedolizumab therapy”

Round 2

Reviewer 1 Report

Comments and Suggestions for Authors

The authors imporved the manuscript following the recommendations. Still, some minor changes may be added.

In the abstract, that is now with more data, the authors may move the sentence the administration of medication to the Methods. Also, in lines 32 and 33, they may avoid the repetition "with a median level of" and use the same topic as in line 37 (xx vs. xx). Avoid using abbreviated words not explained before (see line 39 - the list of inflammation markers may be erased).

Verify the use of abbreviated words in the manuscript (see lines 67 and 95, twice CRP was explained).

In the Legend of each table, the abbreviations must be explained.

The discussion section was improved with some references, but the conclusions may be rewritten to be shorter and concise in the essential results of the study.

Comments on the Quality of English Language

No comments.

Author Response

Thank you to the reviewers for the comments. We believe we adequately responded to the minor comments. However, we did not understand what the reviewer means by "In the Legend of each table, the abbreviations must be explained". 

Reviewer 2 Report

Comments and Suggestions for Authors

No more comments 

Author Response

The reviewer had no further comments.